# Changes in tannin and saponin components during co-composting of *Camellia oleifera* Abel shell and seed cake

**Jinping Zhang** [ORCID] *, **Yue Ying, Xuebin Li, Xiaohua Yao**

Research Institute of Subtropical Forestry Chinese Academy of Forestry, Hangzhou, Zhejiang, China

* jinpingzhang@126.com

**Data Availability Statement:** All relevant data are within the paper and its Supporting Information files.

**Funding:** This work was financially supported by the Provincial Department of Science and

## Abstract

This study investigated the co-compost product of the shell and seed cake of *Camellia oleifera* Abel, which is a small evergreen tree in the family Theaceae. Tannin and saponin contents in compost samples at different time-points and their compositional changes, as well as their relationships with nutrients and compost maturity, were analyzed using an UPLC-triple-TOF/MS system. Our results showed that tannins in the compost samples mainly consisted of 11 phenolic acid compounds, of which four small-molecule phenolic acid compounds were found in low contents. Saponins mainly consisted of five saponin aglycones (A, B, C, D, E) and four of their derivatives in *C. oleifera*. Microbially secreted enzymes converted the large-molecule phenolic acid compounds into small-molecule compounds and their derivatives, while saponins were decomposed into saponin aglycones. Contents of tannin and saponin had correlations with the C/N ratio, germination index (GI), and the Solvita maturity index. After composting, the content of tannin was reduced to less than 1%, and the content of saponin was not more than 2%. And compost products were safe.

## Introduction

*Camellia oleifera* Abel is an economically important woody plant of southern China having the highest production value among the *Camellia* oil tree species [1]. In southern China, *C. oleifera* is commercially grown in the Yangtze River Basin and Pearl River Basin, where the Hunan, Jiangxi, and Guangxi Provinces together account for 76.2% of China's total area under cultivation. This prized plant is also cultivated in small amounts in some areas in Vietnam, Myanmar, Thailand, and Malaysia, as well as Japan [1–2]. Recent statistics show that the annual production of shells and seed cake of *C. oleifera* has reached approximately 8 million and 4 million tons, respectively. The *C. oleifera* shell mainly consists of cellulose, hemicellulose, and lignin, which each decompose rather slowly under natural conditions [3]. The seed cake is the residue that remains after the extraction of oil from *C. oleifera* seeds, and its main components are crude protein, crude fat, fiber, saponin, tannin, ash, and caffeine [4]. Much of *C. oleifera* shells and seed cakes are either burned or discarded, representing a large, needless waste of organic

Technology of Zhejiang, China, Grant
NO.2017C02022

**Competing interests:** The authors have declared
that no competing interests exist.

resources. However, when this waste type is untreated and directly used as an agricultural fertilizer, or is simply thrown away, it generally causes soil and environmental pollution.

But the co-composting of the *C. oleifera* shells and seed cakes enables biological decomposition to proceed in a controlled manner. Due to the heat produced during decomposition, their raw materials are rendered harmless and become stabilized, which can benefit the growth of plants that are fertilized with the co-compost product. Both the physical and chemical properties of the products after decomposition typically differ greatly from those of the initial raw materials (US Composting Council). The *C. oleifera* shell and seed cake contain tannins and saponins in certain amounts. Tannins are polyphenolic compounds that can inhibit microbial growth and resist biological decomposition [5], and *C. oleifera* saponins are pentacyclic triterpenes that are highly toxic to ectotherms. Low concentrations of *C. oleifera* saponins can promote plant growth [6] and may also control pests to a certain extent, including the early instars of *Pieris rapae* (L.) larvae and *Plutella xylostella* (L.) larvae [7]. During the composting process, the temperature of *C. oleifera* shell—seed cake mixture would heat up, and lots of microorganisms that degraded organic matter would grow and reproduce. For instance, *Aspergillus niger* and *Paecilomyces varioti*, they could degrade tannins by hydrolyzing ester and carboxylic acid bonds[8–9]. In addition, glycosidic bond and the ester bonds of saponin would be broken because of changes of pH of substrate. However, no such researches have been reported yet. Hence, studying the dynamics of changed tannin and saponin components during the co-composting of *C. oleifera* shell and seed cake is important if this waste product is to be used sustainably on a large scale.

Here, *C. oleifera* shell and seed cake were used as raw materials in a co-composting experiment. Their tannin and saponin contents were determined by UV spectrophotometry and gravimetric methods, respectively. An ultra-performance liquid chromatography-triple-time-of-flight/mass spectrometry system (UPLC-triple-TOF/MS) was used to analyze changes in the chemical composition and contents of the phenols and saponins in the compost product at different composting time-points. These findings provide a timely basis for the widespread application of the co-compost product of *C. oleifera* shell and seed cake.

## Materials and methods

### Compost materials and preparation

The *C. oleifera* shells were obtained from the Dongfanghong Forest Farm in Jinhua City, and its seed cakes from the Kangneng Tea Oil Co., Ltd., in Tiantai City (both in Zhejiang Province, China). The seed cake was formed after mechanical extraction of the oil from *C. oleifera* seeds, and further oil extraction by No. 6 solvent oil, which primarily consists of alkanes. Effective microorganisms (EMs) were obtained from the Henan Nanhua Qianmu Biotechnology Co., Ltd. in Henan Province, China, whose main components were Bacillus, Lactobacillus, Bifidobacterium, yeast, photosynthetic bacteria, acetic acid bacteria, Actinobacillus, and other original species. Except the addition of EM, laboratory-selected tannins-degrading microorganism agents (*Aspergillus awamori*) and saponin-degrading microorganism agents (*Bacillus amyloliquefaciens* and *Meyerozyma guilliermondii*) were also added. For experiment and collection locations no specific permits were required for the described field studies because the whole experiment process did not involve endangered or protected plant species or privately-owned locations. Basic properties of the co-composting raw materials are summarized Table 1.

### Composting process

The composting experiment was conducted at the Research Institute of Subtropical Forestry, Chinese Academy of Forestry in Fuyang District, Hangzhou City, in Zhejiang Province, from

**Table 1. Basic properties of the co-composting raw materials.**

|  | TOC (%) | TN (%) | C/N | TP (%) | TK (%) | Tannin (%) | Saponin (%) |
|---|---|---|---|---|---|---|---|
| *C. oleifera* shell | 48.6 | 0.42 | 116.00 | 0.0169 | 0.854 | 2.26 | 4.80 |
| *C. oleifera* seed cake | 47.8 | 1.22 | 39.18 | 0.161 | 0.929 | 1.03 | 16.35 |

TN: total nitrogen; TP: total phosphorus; TK: total potassium.

February to April 2019. The experiment took place in an insulated and well-ventilated eco-composter (outer dimensions: 73 cm × 115 cm × 80 cm; volume = 220 L; manufacturer: Biolan, Finland). The dry mass ratio of *C. oleifera* shell (≤ 8 mm) and seed cake (≤ 3 mm) was 4:1. Urea was added to adjust the initial carbon-to-nitrogen ratio (C/N) to 30; water was added to adjust the initial moisture content to 55% (w/w); the EMs (weighing 3% of the dry mass of *C. oleifera* shell and seed cake) were also added. And tannins-degrading microorganism agents (*Aspergillus awamori*) and saponin-degrading microorganism agents (*Bacillus amyloliquefaciens* and *Meyerozyma guilliermondii*) were added in an amount of 1% of the dry mass of *C. oleifera* shell and seed cake, respectively. The raw materials were mixed well and deposited into the eco-composter to begin the composting experiment. After composting began, the temperature of the upper, middle, and lower parts of the compost pile and the ambient temperature were recorded daily at 3:00 pm. Once every 5 days, the compost was turned until the temperature of the compost core matched the room temperature. Samples (each weighing 500 g) were taken on the 0th, 20th, 40th, 60th, and 76th day of composting and were stored at 0°C for later use.

## Analytical methods

For each sample, its tannin content was analyzed via spectrophotometry after extraction with a methanol-water mixture (1:1, v/v) [10]. The saponins were determined according to previous methods [11].

Total organic carbon (TOC), total nitrogen (TN), total phosphorus (TP), total potassium (TK), $NO_3^-$-N, and $NH_4^+$-N were determined following the methodology of Meng et al.[12]. The germination index (GI) was determined as described by Zhang et al.[13], and the Solvita maturity index was determined according to the "Guide to Solvita testing for compost maturity index" (Woods End Research, 2002).

## Analysis using the UPLC-triple-TOF/MS system

Fresh samples of compost, each weighing exactly 10.0 g, were individually placed in a 200-mL conical flask. One hundred milliliters of 80% methanol were added to each flask, which was followed by ultrasonic extraction in a cold-water bath for 30 min (power = 500 W; frequency = 40 kHz). The flasks were centrifuged at 10 000 rpm for 20 min, and their supernatants obtained for analysis using the UPLC-triple-TOF/MS system. The processing of each of the above samples was replicated 3 times.

**Liquid Chromatography (LC).** For all the chromatographic experiments, a Waters UPLC (Waters Corp., Milford, MA, USA), Agilent ZORBAX-SB C18 (100 mm × 4.6 mm i.d: 1.8 μm) was used, for which the mobile phases were 0.1% formic acid-water (A) and 0.1% formic acid-acetonitrile (B). The linear gradient pro-gram parameters were as follows: 0/5, 2/5, 25/50, 33/95 (min/B%); sample injection volume = 10 μL; column oven temperature = 30°C; flow rate = 0.8 mL min$^{-1}$; the UV detector was set to 254 nm.

**Mass Spectrometry (MS).**　For this, an AB TripleTOF 5600$^{plus}$ system (AB SCIEX, Framingham, USA) was used, with these optimal MS conditions applied: scan range = 100–2000 *m/z*; for negative ion mode: source voltage = –4.5 kV and source temperature = 550˚C; for positive ion mode: source voltage = +5.5 kV and source temperature = 600˚C. The pressure of gas 1 (air) and gas 2 (air) was set to 50 psi; the curtain gas ($N_2$) pressure was set to 35 psi. The maximum tolerable error was ± 5 ppm, with a declustering potential (DP) = 100 V and collision energy (CE) = 10 V used. The IDA-based auto-MS2 was performed on the eight most intense metabolite ions in a full scan cycle (1 s). The *m/z* scan range of the precursor ion and product ion were set to 100–2000 Da and 50–2000 Da, respectively. The CE voltage was set to 20 V, 40 V, and 60 V in the positive ESI mode and, conversely, to –20 V, –40 V, and –60 V in the negative ESI mode. Finally, the ion release delay (IRD) used was 67, and the ion release width (IRW) was 25.

The exact mass calibration was performed automatically before each analysis, by using the Automated Calibration Delivery System. Spearman correlations (non-parametric) of the data were carried out in SPSS 20.0 (IBM Co., Armonk, NY, USA).

## Results and discussion

### Changes in contents and components of tannin and saponin during composting

As shown in Fig 1 and Tables 2 and 3, both the tannin and saponin contents continued to decrease during the composting. The decomposition rate of tannins on the 20th, 40th, 60th, and 76th day of composting was 17.27%, 31.48%, 44.88%, and 63.93%, respectively. In the first 20 days of composting, tannins and proteins combined to form a precipitate. At the same time, tannins suppressed the activity of some thermophilic microbes [5], and the microbial decomposition of organic compounds was hindered; this caused temperatures to rise slowly and the decomposition of tannins to proceed slowly during this initial period. But from the 20th to 60th day of composting, microbes evidently became adapted to the inhibitive effects of the tannins. High microbial activity caused the organic matter to decompose rapidly, producing a large amount of heat. The compost pile then entered a high-temperature stage in which the tannins decomposed quickly under the dual effects of microbes and high temperature [14]. The decomposition rate of saponins on the 20th, 40th, 60th, and 76th day of composting was 37.87%, 62.01%, 72.86%, and 86.84%, respectively. The saponins decomposed continuously throughout the experiment's heating phase, high-temperature stage, and cooling periods. As Table 6 shows, the tannin and saponin contents were negatively correlated with the number of days of composting.

As Table 2 shown, in both the starting raw materials and final compost product, the tannins mainly consisted of 11 phenolic acid compounds, and their respective chemical formulas was shown in Table 4. The content of 3,4-dihydroxybenzoic acid, 4-hydroxy-benzoic acid, 7-hydroxy-5-methoxy-2-oxo-2H-chromene-3,6-dicarboxylic acid, p-hydroxybenzaldehyde, 4'-O-(β-D-glucopyranosyl)-3,3',4'-tri-O-methylellagic acid, vitexin, 3-O-methylellagic acid, 3,3'-di-O-methyl ellagic acid, kaempferol, and 3,3',4-tri-O-methylellagic acid in them declined with composting process. The ellagic acid content declined in the first 40 days of composting but increased on its 60th day. This was due to the hydrolysis of 3-O-methylellagic acid, 3,3'-di-O-methyl ellagic acid, and 3',4-tri-O-methylellagic acid, which led to the formation of ellagic acid [15] which replenished and increased its content overall. Since some of the ellagic acid was further metabolized by the microbial community, into urolithin [16–18], the ellagic acid content decreased again on the 76th day of composting. The continuous reduction in tannin content was mainly due to the cleavage of the ester or phenolic bonds of tannins catalyzed by

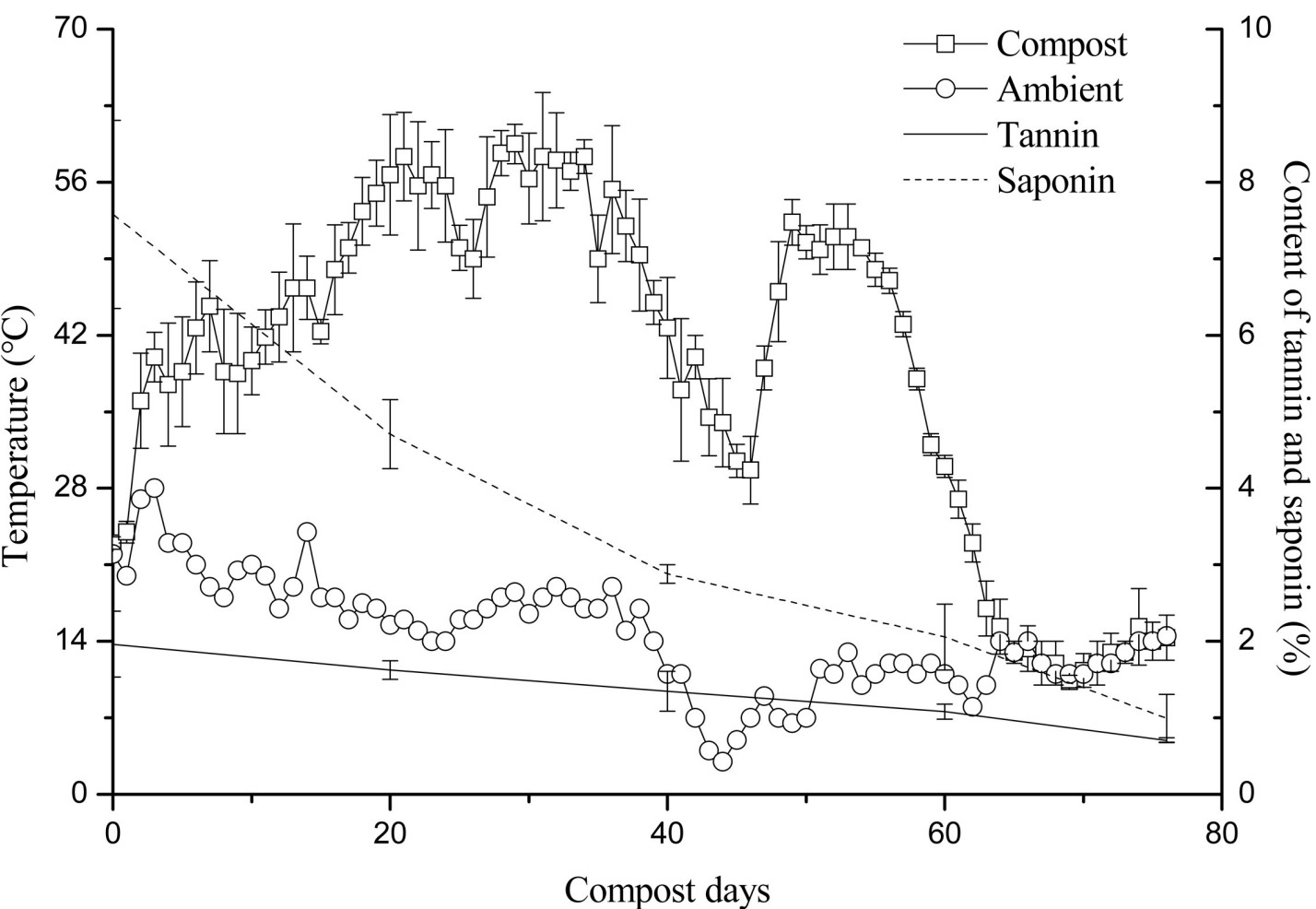

**Fig 1. Changes in tannin and saponin contents during composting of *C. oleifera* shell and seed cake.**

**Table 2. Contents of typical phenolic acids in the compost samples of *C. oleifera* shell and seed cake at different composting time-points.**

| Chemical formula | Chemical name | Response intensity | | | | |
|---|---|---|---|---|---|---|
| | | 0 day | 20 days | 40 days | 60 days | 76 days |
| $C_7H_6O_4$ | 3,4-dihydroxybenzoic acid | 626 | 521 | 483 | 119 | 42 |
| $C_7H_6O_3$ | 4-hydroxy-benzoic acid | 1363 | 316 | 269 | 169 | 143 |
| $C_{12}H_8O_8$ | 7-hydroxy-5-methoxy-2-oxo-2H-chromene-3,6-dicarboxylic acid | 350 | 168 | 151 | 122 | 99 |
| $C_7H_6O_2$ | p-hydroxybenzaldehyde | 355 | 338 | 309 | 425 | 263 |
| $C_{21}H_{18}O_{13}$ | 4'-O-(β-D-glucopyranosyl)-3,3',4'-tri-O-methylellagic acid | 59046 | 917 | 250 | 144 | 148 |
| $C_{21}H_{20}O_{10}$ | vitexin | 119277 | 10451 | 883 | 213 | 251 |
| $C_{14}H_6O_8$ | ellagic acid | 192869 | 63696 | 45799 | 69793 | 67233 |
| $C_{15}H_8O_8$ | 3-O-methylellagic acid | 207028 | 33707 | 30107 | 8540 | 6989 |
| $C_{16}H_{10}O_8$ | 3,3'-di-O-methyl ellagic acid | 116691 | 114925 | 38946 | 2174 | 893 |
| $C_{15}H_{10}O_6$ | kaempferol | 87553 | 6974 | 2622 | 1712 | 886 |
| $C_{17}H_{12}O_8$ | 3,3',4-tri-O-methylellagic acid | 112799 | 81221 | 13895 | 841 | 320 |

The result of relative response was obtained by liquid chromatography (LC).

**Table 3. Contents of typical saponins in compost samples of *C. oleifera* shell and seed cake at different composting time-points.**

| Chemical formula | Chemical name | Response intensity | | | | |
|---|---|---|---|---|---|---|
| | | 0 day | 20 days | 40 days | 60 days | 76 days |
| $C_{53}H_{86}O_{26}$ | desacyl-theasaponin A | 480428 | 4319 | 1360 | 50 | 34 |
| $C_{53}H_{84}O_{25}$ | desacyl-theasaponin E methyl ester | 553167 | 60762 | 12632 | 496 | 538 |
| $C_{43}H_{68}O_{17}$ | Prosapogenol | 409005 | 5642 | 3800 | 52 | 39 |
| $C_{30}H_{50}O_6$ | theasapogenol A | 777130 | 24819 | 76356 | 2208 | 2194 |
| $C_{30}H_{48}O_6$ | camelliagenin D | 1373366 | 68201 | 232396 | 2101 | 1928 |
| $C_{30}H_{50}O_5$ | camelliagenin C | 345046 | 25071 | 51791 | 6425 | 4844 |
| $C_{29}H_{46}O_5$ | camelliagenin D | 490142 | 124771 | 763485 | 66368 | 27814 |
| $C_{30}H_{48}O_5$ | camelliagenin B | 822892 | 65810 | 82132 | 3827 | 3633 |
| $C_{30}H_{50}O_4$ | camelliagenin A | 165745 | 22051 | 26608 | 4131 | 3280 |

The result of relative response was obtained by liquid chromatography (LC).

microbially secreted tannase, respectively forming phenolic acids or polyols. Under the enzymatic actions of polyphenol oxidase and decarboxylase, these compounds were further decomposed into phloroglucinol and resorcinol, and eventually into 3-hydroxy-5-oxo-hexanoic acid, 5-oxo-6-methyl hexanoate, and other small molecules. Some of these molecules were utilized as carbon sources, while some were converted to phenol derivatives [19–20]. The raw materials of the co-compost of *C. oleifera* shell and seed cake had low contents of these four phenolic acid compounds: 3,4-dihydroxybenzoic acid, 4-hydroxy-benzoic acid, 7-hydroxy-5-methoxy-2-oxo-2H-chromene-3,6-dicarboxylic acid, and p-hydroxybenzaldehyde. By contrast, the initial contents of 4'-O-(β-D-glucopyranosyl)-3,3',4'-tri-O-methylellagic acid, vitexin, ellagic acid, 3-O-methylellagic acid, 3,3'-di-O-methyl ellagic acid, kaempferol, and 3,3',4-tri-O-methylellagic acid were all higher than those four compounds.

*C. oleifera* saponins are pentacyclic triterpenes and they contain ligands, glycosyl groups, and small-molecule organic acids. The organic acids consist mostly of angelic acid and acetic acid. Glycosyl groups include glucuronic acid, arabinose, xylose, and galactose [21,22]. Before the composting, saponins in the *C. oleifera* shell and seed cake mainly contained glycosides and glycosyl groups, yet after composting, saponins contained glycosyl groups chiefly. In particular, the content of the camelliagenin D derivative ($C_{29}H_{46}O_5$) was found to be highest after composting, a result consistent with the findings by Lin et al. [22]. *C. oleifera* saponins in the compost samples mainly included glycoside molecules, including desacyl-theasaponin A ($C_{53}H_{86}O_{26}$), desacyl-theasaponin E methyl ester ($C_{53}H_{84}O_{25}$), and prosapogeno ($C_{43}H_{68}O_{17}$), and aglycone molecules, including theasapogenol A ($C_{30}H_{50}O_6$), camelliagenin D ($C_{30}H_{48}O_6$), camelliagenin C ($C_{30}H_{50}O_5$), camelliagenin D derivative ($C_{29}H_{46}O_5$), camelliagenin B ($C_{30}H_{48}O_5$), and camelliagenin A ($C_{30}H_{50}O_4$) (Table 3 and Table 5). The content of glycoside molecules continued to decline as composting proceeded, whereas the aglycone molecules were already reduced by the 20th day, but increased on the 40th day, yet decreased again after the 60th day of composting. This pattern was driven by the hydrolysis of the glycosidic bond of *C. oleifera* saponins by acids, bases, or enzymes into sugars and aglycones. According to Hu et al. [23], under acidic conditions, the glycosidic bond at the 3-position was broken, while under alkaline conditions, ester bonds at the 16-, 18-, 21-, and 22-position were broken. Saponin aglycones and sugars were mineralized under the actions of microbial enzymes and were eventually converted to $CO_2$ and water. The increase in saponin aglycone content evident on the 40th day of composting was likely caused by the decomposition of glycoside molecules to saponin aglycones.

**Table 4. Chemical names of typical phenolic acid compounds in the compost samples (of *C. oleifera* shell and seed cake) and their respective chemical formulas.**

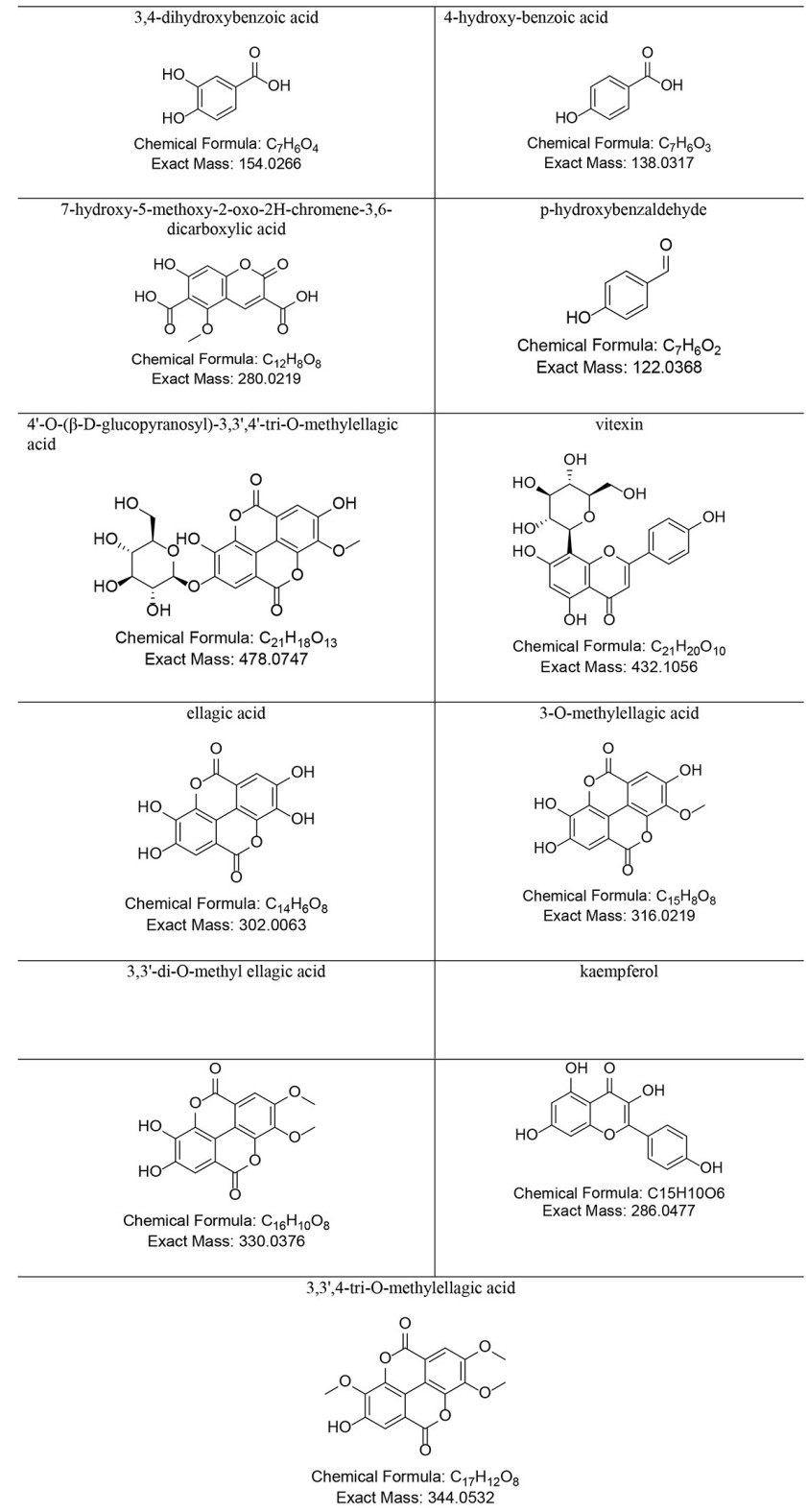

| 3,4-dihydroxybenzoic acid | 4-hydroxy-benzoic acid |
|---|---|
| Chemical Formula: $C_7H_6O_4$<br>Exact Mass: 154.0266 | Chemical Formula: $C_7H_6O_3$<br>Exact Mass: 138.0317 |
| 7-hydroxy-5-methoxy-2-oxo-2H-chromene-3,6-dicarboxylic acid | p-hydroxybenzaldehyde |
| Chemical Formula: $C_{12}H_8O_8$<br>Exact Mass: 280.0219 | Chemical Formula: $C_7H_6O_2$<br>Exact Mass: 122.0368 |
| 4'-O-(β-D-glucopyranosyl)-3,3',4'-tri-O-methylellagic acid | vitexin |
| Chemical Formula: $C_{21}H_{18}O_{13}$<br>Exact Mass: 478.0747 | Chemical Formula: $C_{21}H_{20}O_{10}$<br>Exact Mass: 432.1056 |
| ellagic acid | 3-O-methylellagic acid |
| Chemical Formula: $C_{14}H_6O_8$<br>Exact Mass: 302.0063 | Chemical Formula: $C_{15}H_8O_8$<br>Exact Mass: 316.0219 |
| 3,3'-di-O-methyl ellagic acid | kaempferol |
| Chemical Formula: $C_{16}H_{10}O_8$<br>Exact Mass: 330.0376 | Chemical Formula: C15H10O6<br>Exact Mass: 286.0477 |
| 3,3',4-tri-O-methylellagic acid | |
| Chemical Formula: $C_{17}H_{12}O_8$<br>Exact Mass: 344.0532 | |

**Table 5. Chemical names of typical saponins in compost samples (of *C. oleifera* shell and seed cake) and their respective chemical formulas.**

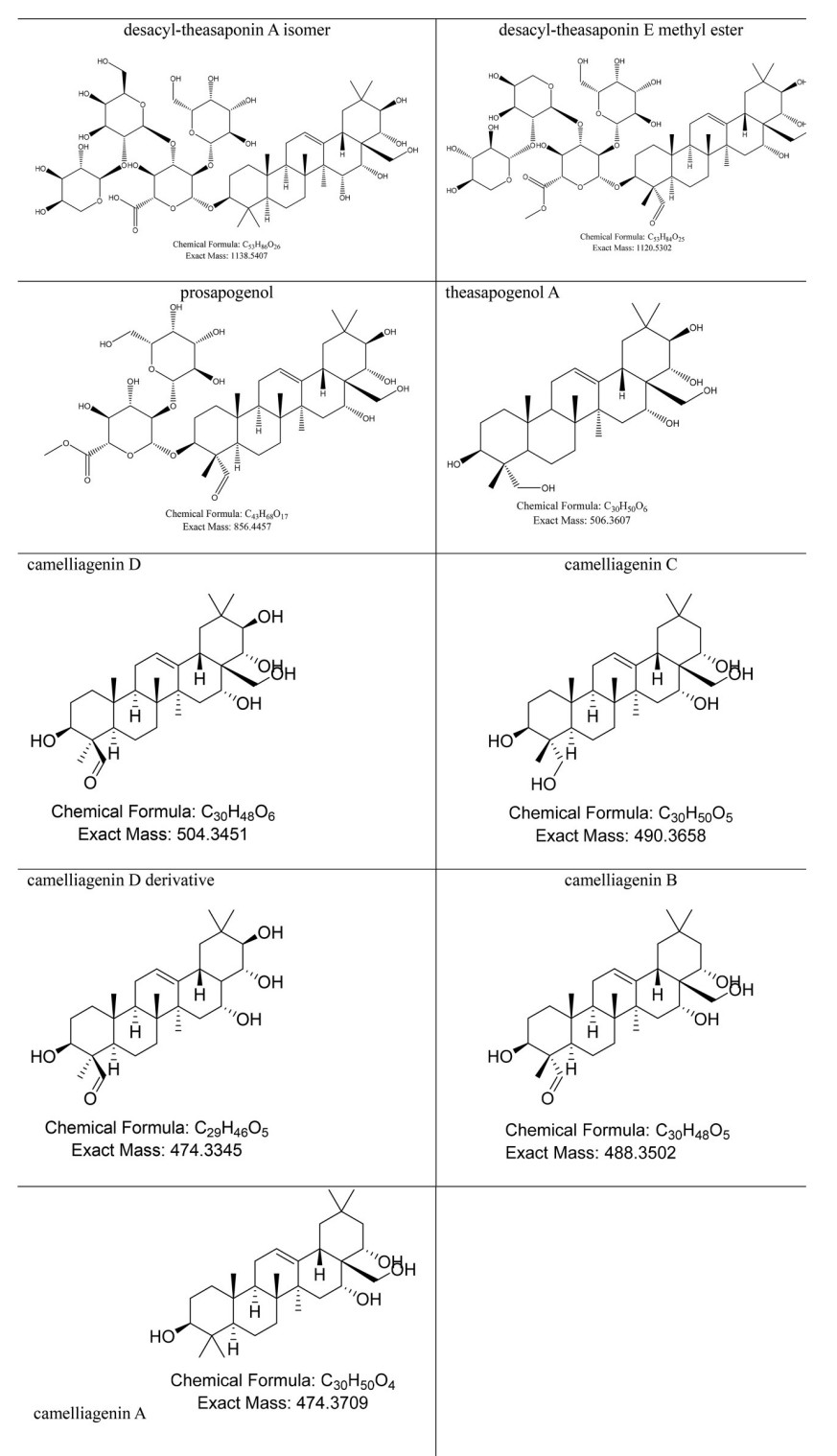

**Table 6. Spearman correlations among physical and chemical parameters during composting of *C. oleifera* shell and seed cake.**

| | Day | Tannin | Saponin | TN | TP | TK | NO$_3^-$-N | NH$_4^+$-N | Total nutrient content |
|---|---|---|---|---|---|---|---|---|---|
| Day | 1 | −0.996** | −0.973** | 0.981** | 0.979** | 0.958* | 0.902* | −0.919* | 0.992** |
| Tannin (%) | | 1 | 0.967** | −0.958* | −0.958* | −.0966** | −0.917* | 0.906* | −0.979** |
| Saponin (%) | | | 1 | −0.957* | −0.977** | −0.983** | −0.789 | 0.984** | −0.984** |
| TN | | | | 1 | 0.995** | 0.911* | 0.846 | −0.918* | 0.992** |
| TP | | | | | 1 | 0.936* | 0.817 | −0.951* | 0.996** |
| TK | | | | | | 1 | 0.814 | −0.958* | 0.956* |
| NO$_3^-$-N | | | | | | | 1 | −0.670 | 0.851 |
| NH$_4^+$-N | | | | | | | | 1 | −0.949* |
| Total nutrient content | | | | | | | | | 1 |

TN: total nitrogen; TP: total phosphorus; TK: total potassium.

### Relationships between tannin and saponin contents and nutrients

As shown in Figs 2 and 3, the TN, TP, TK, NO$_3^-$-N, and total nutrient contents increased with, and were all positively correlated with, the duration of composting time (Table 6). When

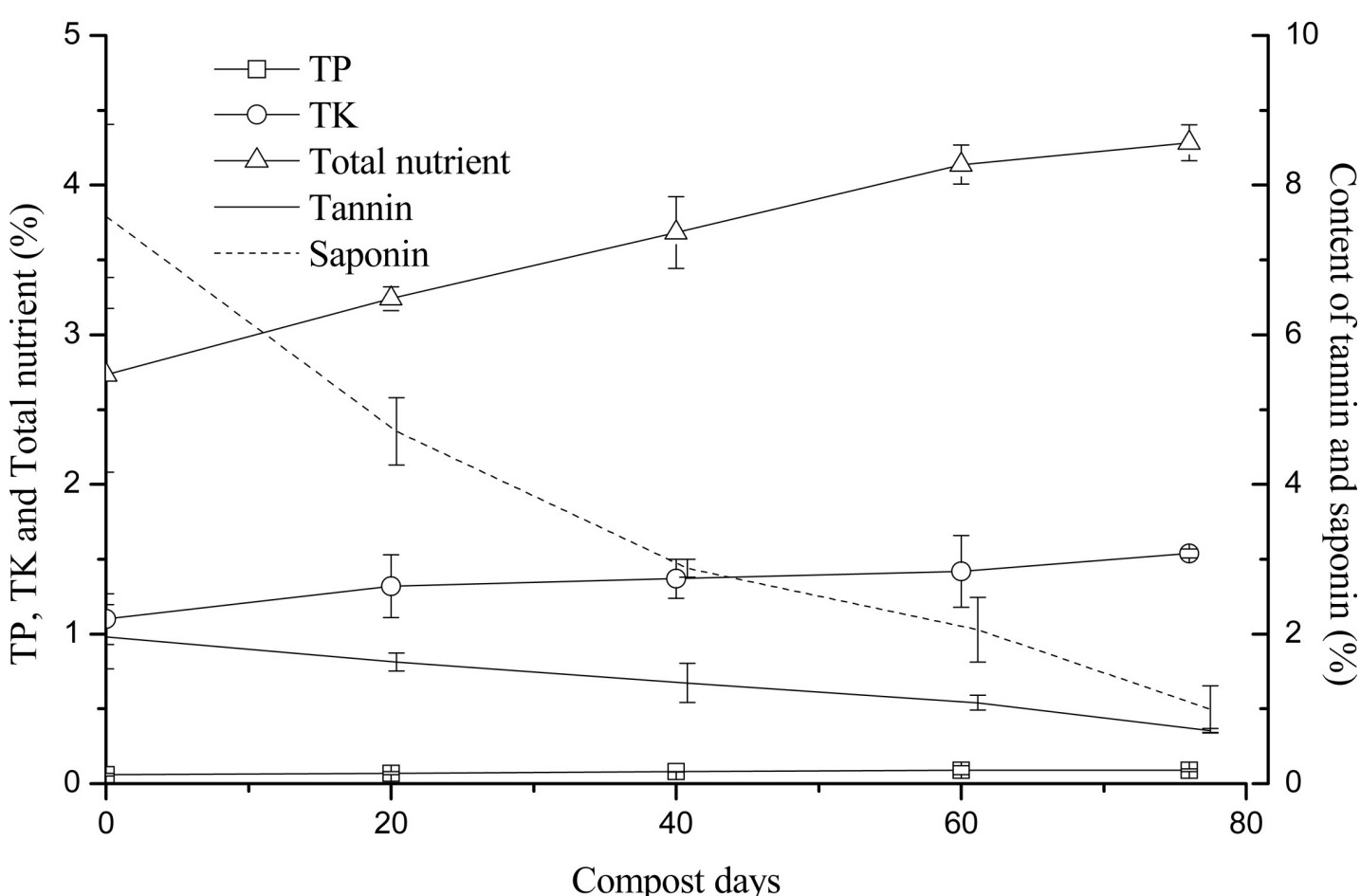

**Fig 2. Relationships between the tannin and saponin contents and the TN, TP, TK, and total nutrient contents.** TN: total nitrogen, TP: total phosphorus, TK: total potassium.

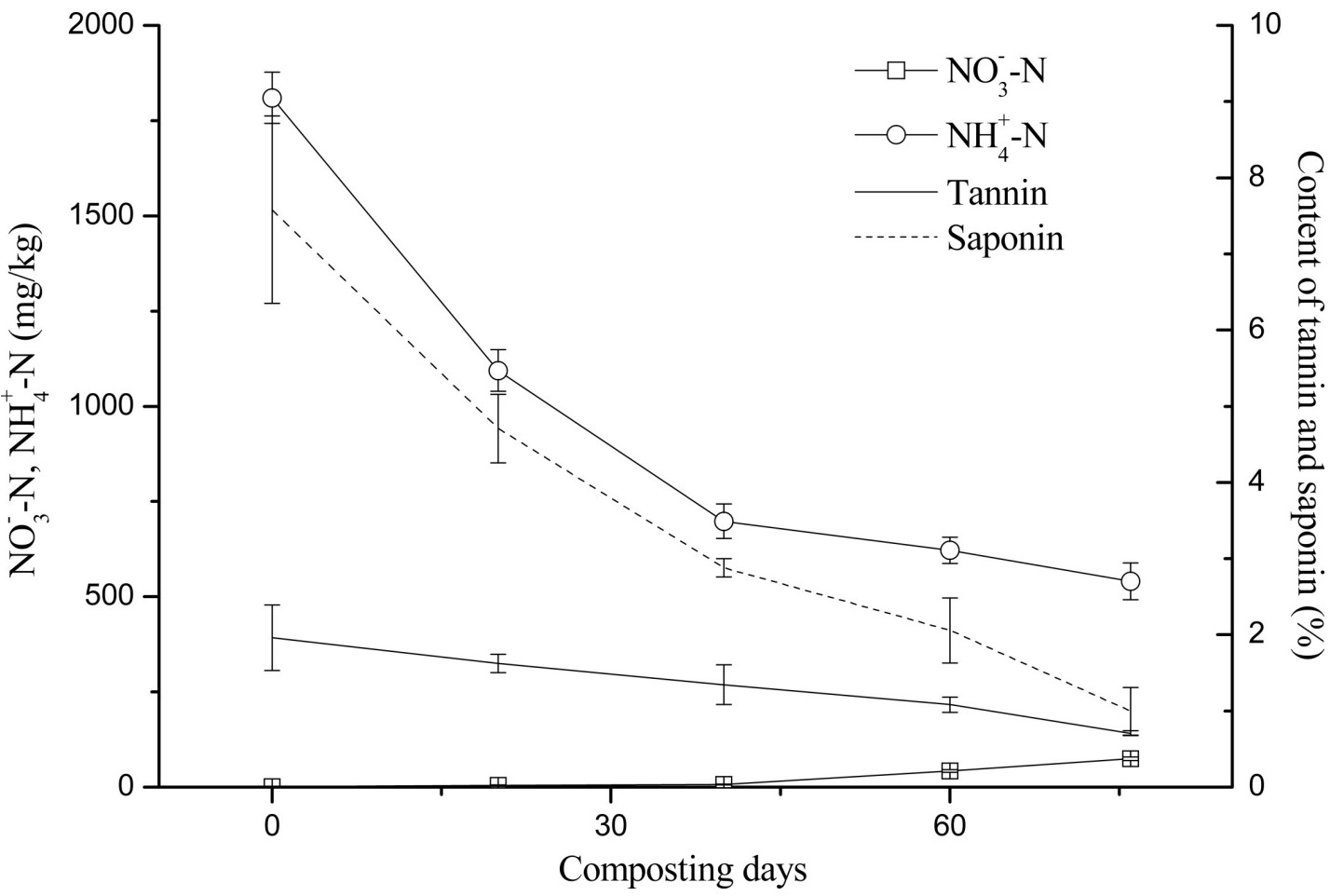

**Fig 3. Relationships between the tannin and saponin contents and NO3⁻-N and NH4⁺-N.**

composting began, the TN content was 1.27 ± 0.02 g/kg, which increased to 12.71 ± 0.22 g/kg on the 20th day, 14.92 ± 0.03 g/kg on the 40th day, 18.47 ± 0.06 g/kg on the 60th day, and 22.15 ± 0.04 g/kg, and 22.17 ± 0.16 g/kg on the 76th day. This pronounced increase was likely due to the net loss of dry matter, specifically in terms of $CO_2$ and evaporative loss of water, as driven by the heat produced during the oxidation of organic matter [24]. The TN, TP, TK, and total nutrient contents were all negatively correlated with the tannin and saponin contents (Table 6). During the composting process, the TP and TK contents increased slightly (Fig 2), possibly because of the net loss of dry mass.

Because $NO_3^-$-N is the main source of nitrogen available for uptake by most plants [25], for a given compost product the higher its $NO_3^-$-N content, the higher will be its fertilizer efficiency. As Fig 3 shows, the $NO_3^-$-N content changed little over the first 40 days of composting but it increased rapidly during days 60–76. Some of the $NH_4^+$-N formed during microbial deamination was probably volatilized as ammonia gas, while some would have been converted to $NO_3^-$-N by ammonia-oxidizing microbes [26]. In the early stage of composting, when both its pH and temperature were relatively high, ammonia volatilization caused the rapid reduction of $NH_4^+$-N in the compost pile. By the end of the composting period, the $NH_4^+$-N content had been reduced to less than one-third of its starting value (i.e., at day zero of the experiment), whereas the $NO_3^-$-N content had risen to more than 40 times its initial value. The tannin content was

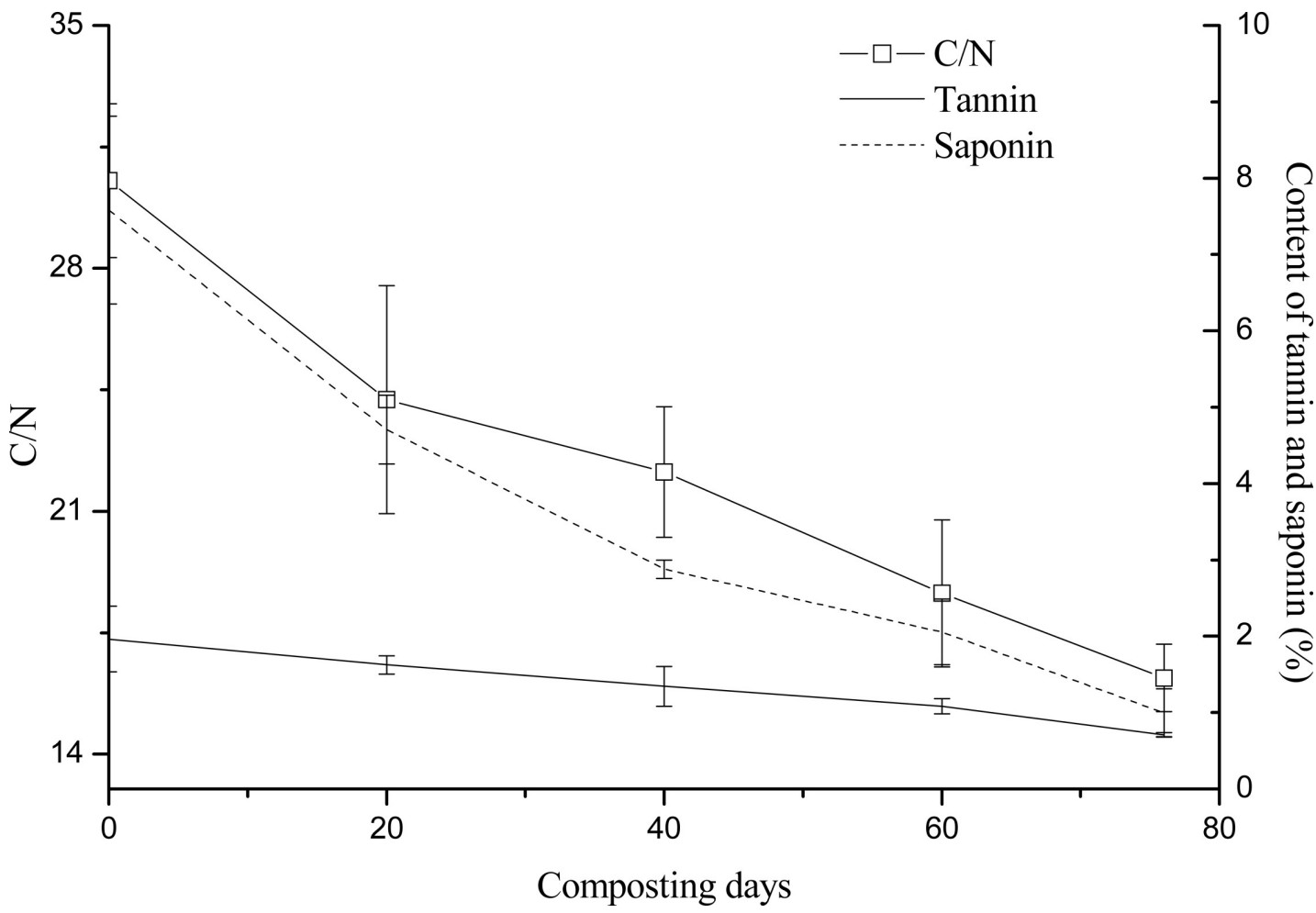

**Fig 4. Relationships between tannin and saponin contents and the C/N ratio.**

negatively correlated with $NO_3^-$-N yet positively so with $NH_4^+$-N. The saponin content was also positively correlated with $NH_4^+$-N, but negatively correlated with $NO_3^-$-N (Table 6).

## Relationships between tannin and saponin contents and maturity

Compost maturity is related to the safety and stability of the compost product, so it serves as an important indicator for evaluating the quality of the compost product. In this study, the C/N ratio, GI, and Solvita maturity index were relied upon to evaluate maturity, and their relationships with tannin and saponin contents were investigated. The initial C/N of the raw compost materials was 30, but it fell to less than 20 on the 60[th] day of composting. On the 76[th] day, the compost temperature was now close to room temperature, so the C/N remained relatively stable at 16.2 (Fig 4). At this time, T = 0.53, a result consistent with by Morel et al. [27] finding that the compost reached its maturity when T $[(C/N)_{final} / (C/N)_{initial}] < 0.6$. The GI refers to the toxicity of the compost product to plants; when GI > 100%, the product is considered to nontoxic. As Fig 5 shown, the compost product attained a nontoxic status (>100%) on the 40[th] day. The Solvita maturity index is a widely accepted index easily obtained through simple testing. From the start of the composting, this index followed a general trend of continual increase, in that it reached values of 3, 5, 7, and 8, on the 20[th], 40[th], 60[th], and 76[th] days of

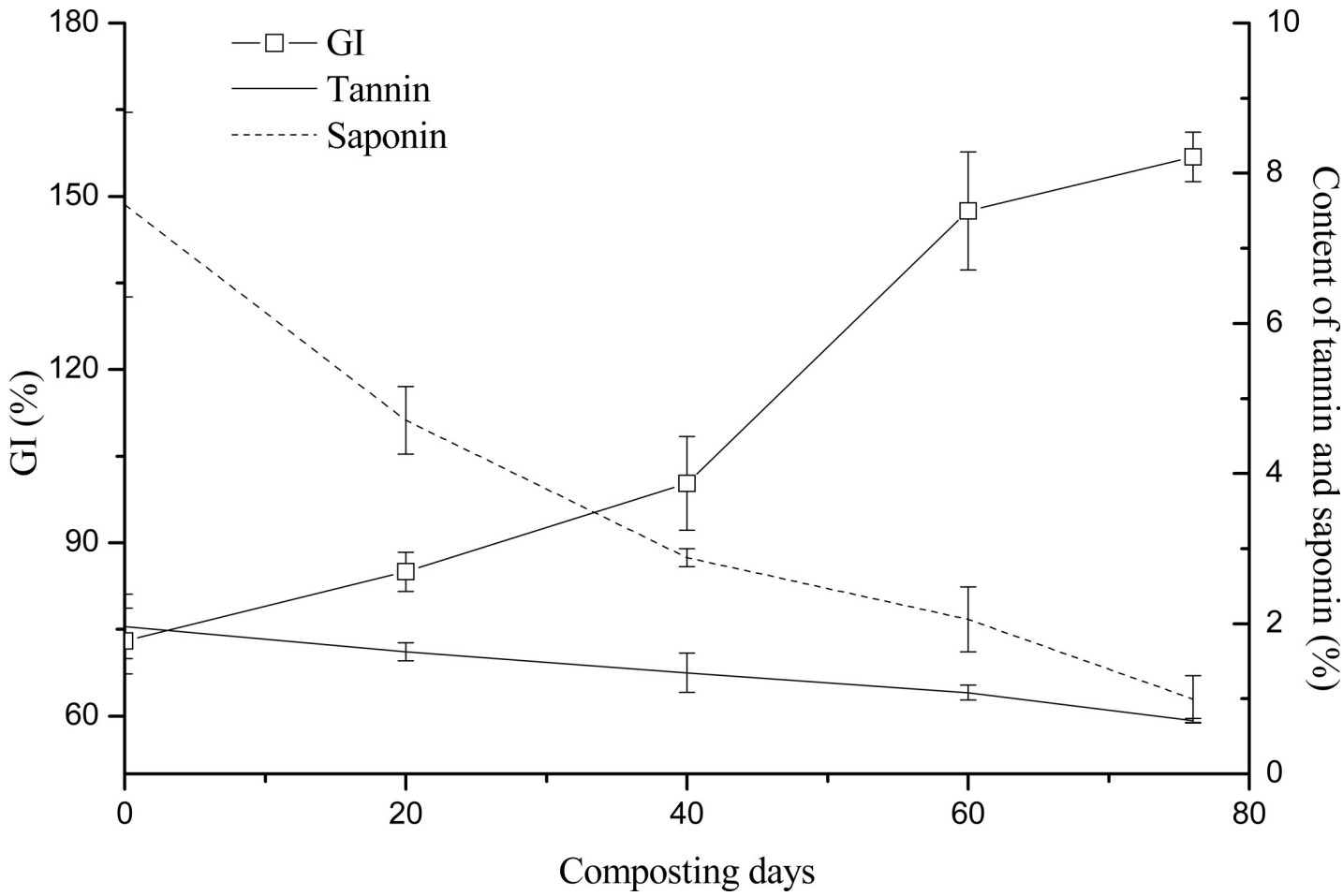

**Fig 5. Relationships between tannin and saponin contents and the GI.**

**Table 7. Solvita maturity index during the composting of *C. oleifera* shell and seed cake.**

| Days | 20 | 40 | 60 | 76 |
|---|---|---|---|---|
| Solvita maturity index | 3 | 5 | 7 | 8 |

**Table 8. Correlations of tannin and saponin contents with the C/N ratio, GI, and Solvita maturity index during composting of *C. oleifera* shell and seed cake.**

|  | Day | Tannin (%) | Saponin (%) | C/N | GI | Solvita maturity index |
|---|---|---|---|---|---|---|
| Day | 1 | −0.996** | −0.973** | −0.983** | 0.968** | 0.991** |
| Tannin (%) |  | 1 | 0.967** | 0.980** | −0.956* | −0.981** |
| Saponin (%) |  |  | 1 | 0.989** | −0.896* | −0.992** |
| C/N |  |  |  | 1 | −0.932* | −0.995** |
| GI |  |  |  |  | 1 | 0.944* |
| Solvita |  |  |  |  |  | 1 |

* Correlation is significant at 0.05

** Correlation is significant at 0.01

composting, respectively (Table 7). According to the "Guide to Solvita testing for compost maturity index" (Woods End Research, 2002), when the Solvita maturity index is greater than 6, the compost is considered to have matured. Therefore, we may infer that on the 60[th] day of composting, the compost of *C. oleifera* shells and seed cake had reached its state of maturity. We also found the tannin and saponin contents positively correlated with C/N but negatively correlated with both GI and Solvita maturity index (Table 8). In view of the C/N, GI, and Solvita maturity index results, we are confident that the compost had matured on the 60[th] day of composting, and that it is more appropriate to have tannin and saponin contents respectively of $\leq$ 1% and $\leq$ 2% in the compost product.

## Conclusion

Tannins and saponins of *C. oleifera* shell and seed cake can be degraded, and the content of $NO_3$-N can be increased by composting technology with 60 days. Thus, qualified compost products that are beneficial to plant growth can be obtained. In the experimental compost samples, the identified tannins mainly consisted of 11 phenolic acid compounds, but four of these were small-molecule phenolic acids relatively low in content. The saponins identified mainly consisted of five saponin aglycones of *C. oleifera* (named A, B, C, D, and E) and four of their derivatives. With the progress of composting, the content of tannins and saponins gradually decreased to a safe range. Because microbially secreted enzymes converted most large-molecule phenolic acid compounds to small-molecule compounds and their derivatives, and saponins decomposed to saponin aglycones. In addition, the tannin and saponin contents were inversely related to the TN, TP, TK, GI, Solvita maturity index and total nutrient contents, and the negative correlation with Solvita maturity index was significant. Yet the tannin and saponin contents were significantly positively correlated with C/N and content of $NO_3$-N.

## Acknowledgments

The authors would like to thank Zhiwei Ge (Analysis Center of Agrobiology and Environmental Sciences, Zhejiang University) for the technical assistance.

## Author Contributions

**Data curation:** Yue Ying.

**Methodology:** Xuebin Li.

**Resources:** Xiaohua Yao.

**Software:** Yue Ying.

**Writing – original draft:** Jinping Zhang.

**Writing – review & editing:** Jinping Zhang.

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
