## [Decision Letter · Decision Letter 0]

28 Jan 2020

PONE-D-19-36037

Changes in tannin and saponin components during co-composting of Camellia oleifera Abel shell and seed cake

PLOS ONE

Dear Mrs Zhang,

Thank you for submitting your manuscript to PLOS ONE. After careful consideration, we feel that it has merit but does not fully meet PLOS ONE’s publication criteria as it currently stands. Therefore, we invite you to submit a revised version of the manuscript that addresses the points raised during the review process.

Both reviewers have found the manuscript topic interesting; however, they addressed several questions to the authors which should be elaborated before the manuscript can move to further processing.

We would appreciate receiving your revised manuscript by Mar 13 2020 11:59PM. To enhance the reproducibility of your results, we recommend that if applicable you deposit your laboratory protocols in protocols.io, where a protocol can be assigned its own identifier (DOI) such that it can be cited independently in the future. For instructions see: http://journals.plos.org/plosone/s/submission-guidelines#loc-laboratory-protocols

We look forward to receiving your revised manuscript.

Kind regards,

Branislav T. Šiler, Ph.D.

Academic Editor

PLOS ONE

Journal Requirements:

"The authors are grateful for the financial support from the Provincial Department of Science and Technology of

Zhejiang, China (Grant No. 2017C02022)."

NO

Reviewers' comments:

Reviewer's Responses to Questions

**Comments to the Author**

1. Is the manuscript technically sound, and do the data support the conclusions?

Reviewer #1: Partly

Reviewer #2: Yes

2. Has the statistical analysis been performed appropriately and rigorously? 

Reviewer #1: Yes

Reviewer #2: Yes

3. Have the authors made all data underlying the findings in their manuscript fully available?

Reviewer #1: Yes

Reviewer #2: Yes

4. Is the manuscript presented in an intelligible fashion and written in standard English?

Reviewer #1: Yes

Reviewer #2: Yes

5. Review Comments to the Author

Reviewer #1: The authors gave a intresting work focus on the degradation of tannin and saponin during composting of the shell and seed cake of Camellia oleifera Abel. Which make a lot of sence for making resue of the Camellia oleifera Abel wastes without any harmful effect caused by tannin and saponin. However, the manuscript in current version was not good enough to publish.

First is introduction. The authors gave enough references to support the idea that the shell and seed cake of Camellia oleifera Abel should be and can be treated by composting, but why the tannin and saponin could be degraded ? More details about the research progress about it should be given, and the challenges should be pointed out based these reference.

Second is the sturcture of the manuscript. For as a composting experiment, its effect should be considered based on a certain condition that the composting was normally proceeded. And then, the changes of nutrients, maturity and degradation of tannin and saponin should be described one by one, and then the ralationship were analyzed.

The authors gave a same name and contents for the first two parts in results and discussion part, and the dicussion on the results was not enough. Both of the problem made the results and discussion as a report rather than a research paper.

The last is the strange figures and tables. Both of the figures and tables were not shown as standard ones in any publication.

1) the figure numbers were not consistent with those in the manuscript.

2) the contents of tannin and saponin were shown in nearly all of the figures, which should be discussed in the most related one of them or shown separatedly.

3) the words were shown in different format in various figures.

Reviewer #2: The article deals with co-composting of Camellia oleifera shells and seed cake, which are by-products obtained during the oil extraction from Camellia oleifera. The work provides relevant information on plant nutrition and safety parameters using Camellia oleifera compost, which is interesting from the agronomic and environmental point of view. The valorization of oil industry by-products is a hot topic, as they are an environmental concern, and their reuse would improve the sustainability of the oil production system. Therefore, I suggest to accept the article for its publication after some amendments of the text.

Abstract

Replace the text “(...) Camellia oleifera Abel. It was a family of theaceae, small evergreen trees” with “(...) Camellia oleifera Abel, which is a small evergreen tree in the family Theaceae.”

Correct the sentence “were found low in content”: four small-molecule phenolic acids were found in low contents.

Correct the verb tense “it takes 60 days”: it took 60 days.

The sentence “tannin and saponin contents were examined for their correlations (...) repeats a previous sentence in the abstract. It would be simpler and clearer as follows: “Tannin and saponin correlations with the C/N ratio, germination index (GI), and the Solvita maturity

Page 2 and beyond

Check the bibliography format, most of the citations are undistinguished numbers within the text.

Page 4

There is a reference error message (analytical methods).

The year of the reference 8 is wrong: Savolainen, H. (1992). Tannin content of tea and coffee. Journal of Applied Toxicology, 12(3), 191-192.

Typographical error: pro-gram.

Page 5

Table 6 is referred after table 3, therefore, it should be table 4. The rest of table numbers should be corrected accordingly.

Please add a coma and reorganize the sentence for better understanding: “As composting proceeded, the content of every phenolic acid compound declined, except for ellagic acid”.

Page 6

Tables 2 and 3. Is response intensity the relative response in the MS detector? It should be stated as a table footnote.

The term “tea saponins” is confusing, as the article does not deal with tea (Camellia sinensis). I would refer it as C. oleifera saponins or its common name (tea oil camellia, provided that it has been defined previously in the introduction).

There is a confusion with saponin aglycones names. In some parts of the text (e.g. abstract) they are defined as “A, B, C, D and E, while in others (and table 3) as theasapogenol A, camelliagenin A, camelliagenin B, camelliagenin C, camelliagenin D and camelliagenin D derivative. Only one name should be used, and I would suggest using the last ones, as they are more descriptive of the compounds.

Description of saponins presence in compost is repeated twice. There is a first sentence “saponin contents (...) primarily consisted of five saponin aglycones of C. oleifera –A, B, C, D, and E- and four of their derivatives.” and below a second sentence “Tea saponins in the compost samples mainly included (...)”, which agrees with the information in tables 3 and 5. I would suggest replacing the first sentence with the second one. In addition, the number of glycosides/aglycones seems to be wrong in one of the sentences, as the first says “5 aglycones/4 derivatives” and the second describes 6 aglycones and 3 glycosides. Please, verify the number of glycosides/aglycones.

Page 10:

(Table 6) instead of (see Table 6).

Is there a direct relationship between ammoniacal nitrogen or nitrate nitrogen and tannin or saponin contents? Or, are the found correlations just due to their increase/decrease with composting time?

Page 11

Table 6: Definition of TN, TP and TK should be stated as a footnote.

Conclusion

Please check the saponins names (as in abstract and page 6) so they are consistent throughout text and tables. Also, check the number of aglycones and glycosides.

6. PLOS authors have the option to publish the peer review history of their article (what does this mean?). If published, this will include your full peer review and any attached files.

Reviewer #1: No

Reviewer #2: No

---

## [Author Response · Author response to Decision Letter 0]

19 Feb 2020

Responses to Reviewer 1:

Reviewer #1: The authors gave an intresting work focus on the degradation of tannin and saponin during composting of the shell and seed cake of Camellia oleifera Abel. Which make a lot of sence for making resue of the Camellia oleifera Abel wastes without any harmful effect caused by tannin and saponin. However, the manuscript in current version was not good enough to publish.

Response: Thank you for your working on our manuscript. All the points in this comment were already revised in this revision, and we also made some other changes that were highlighted by red color.

First is introduction. The authors gave enough references to support the idea that the shell and seed cake of Camellia oleifera Abel should be and can be treated by composting, but why the tannin and saponin could be degraded? More details about the research progress about it should be given, and the challenges should be pointed out based these references.

Response: Thank you for your suggestions. These were revised in introduction. Firstly, the reasons for the degradation of tannin and saponin were added in the introduction. Secondly, the biggest challenge was experimental design, because there were no references concerning tannin and saponin of C. oleifera shells and seed cakes. This point has also been added in the introduction. And details about the research progress have been explained in detail in the experimental methods (shown in line56-60). 

Second is the sturcture of the manuscript. For as a composting experiment, its effect should be considered based on a certain condition that the composting was normally proceeded. And then, the changes of nutrients, maturity and degradation of tannin and saponin should be described one by one, and then the ralationship were analyzed.

Response: Your suggestion makes some sense. In this manuscript, the results and discussion of changes of tannin and saponin were analyzed strictly in accordance with the composting normal process. However, the changes of nutrients and maturity were not only the certain condition of changes of tannin and saponin but also the results of changes of tannin and saponin. We were aiming to investigate the changes of tannin and saponin in this work. As shown in results and discussion, combined with the changes of nutrients, maturity and degradation of tannin and saponin were analyzed. Thus, it’s difficult and illogical to change the structure in the order you suggested fully. So, we retained the arrangement. 

The authors gave a same name and contents for the first two parts in results and discussion part, and the discussion on the results was not enough. Both of problem made the results and discussion as a report rather than a research paper.

Response: First two parts in results and discussion part analyzed and discussed the changes of tannin and saponin from content and composition, respectively. The titles of the two parts were too similar that it was hard to distinguish, so they were merged. And results and discussion part supplemented the discussion.

The last is the strange figures and tables. Both of the figures and tables were not shown as standard ones in any publication.

1) the figure numbers were not consistent with those in the manuscript.

2) the contents of tannin and saponin were shown in nearly all of the figures, which should be discussed in the most related one of them or shown separately.

3) the words were shown in different format in various figures.

Response: Thank you for your pointing out the problem. All the formatted problems about figures and tables were revised in this revision. The contents of tannin and saponin were shown in all the figures to compare the impact of the same compost parameters on them. In addition, it also enriched the information of the graph. In results and discussion part, the effects of same compost parameters on tannin and saponin were discussed separately.

Responses to Reviewer 2:

Reviewer #2: The article deals with co-composting of Camellia oleifera shells and seed cake, which are by-products obtained during the oil extraction from Camellia oleifera. The work provides relevant information on plant nutrition and safety parameters using Camellia oleifera compost, which is interesting from the agronomic and environmental point of view. The valorization of oil industry by-products is a hot topic, as they are an environmental concern, and their reuse would improve the sustainability of the oil production system. Therefore, I suggest to accept the article for its publication after some amendments of the text.

Response: Thank you for your working on our manuscript. All the points in this comment were already revised in this revision, and we also made some other changes that were highlighted by red color.

Abstract

Replace the text “(...) Camellia oleifera Abel. It was a family of theaceae, small evergreen trees” with “(...) Camellia oleifera Abel, which is a small evergreen tree in the family Theaceae.”

Correct the sentence “were found low in content”: four small-molecule phenolic acids were found in low contents.

Correct the verb tense “it takes 60 days”: it took 60 days.

The sentence “tannin and saponin contents were examined for their correlations (...) repeats a previous sentence in the abstract. It would be simpler and clearer as follows: “Tannin and saponin correlations with the C/N ratio, germination index (GI), and the Solvita maturity

Response: Thank you for your suggestions. These sentences were revised according to your suggestions, and the tense in the whole manuscript was checked (shown in 22-23, 26, 29-30).

Page 2 and beyond

Check the bibliography format, most of the citations are undistinguished numbers within the text.

Response: This format was according to the requirement of PLOS ONE’s bibliography format (for example, “[1]” or “[2-5]”or “[3,7,9]”).

Page 4

There is a reference error message (analytical methods).

The year of the reference 8 is wrong: Savolainen, H. (1992). Tannin content of tea and coffee. Journal of Applied Toxicology, 12(3), 191-192.

Typographical error: pro-gram.

Response: This have been revised, and checked for other bibliographic information. 

Page 5

Table 6 is referred after table 3, therefore, it should be table 4. The rest of table numbers should be corrected accordingly.

Response: This was because Table 6 summarized the correlation analysis results of all indicators of compost and tannin and saponin contents. Hence, Table 6 was referred in here. However, the order of Tables was correct.

Please add a coma and reorganize the sentence for better understanding: “As composting proceeded, the content of every phenolic acid compound declined, except for ellagic acid”.

Response: This sentence had been revised in the text in easy understanding way (shown in 142-146).

Page 6

Tables 2 and 3. Is response intensity the relative response in the MS detector? It should be stated as a table footnote.

Response: No, response intensity is the relative response in the Liquid chromatography (LC), it has been stated as a table footnote.

The term “tea saponins” is confusing, as the article does not deal with tea (Camellia sinensis). I would refer it as C. oleifera saponins or its common name (tea oil camellia, provided that it has been defined previously in the introduction).

Response: The term “tea saponins” were pentacyclic triterpenes and has been explained in detail in 166-167 lines on page 7 in revision. 

There is a confusion with saponin aglycones names. In some parts of the text (e.g. abstract) they are defined as “A, B, C, D and E, while in others (and table 3) as theasapogenol A, camelliagenin A, camelliagenin B, camelliagenin C, camelliagenin D and camelliagenin D derivative. Only one name should be used, and I would suggest using the last ones, as they are more descriptive of the compounds.

Response: In the introduction, A, B, C, D, and E all represented saponin aglycone, but theasapogenol A, camelliagenin A, camelliagenin B, camelliagenin C, camelliagenin D and camelliagenin D derivative, and desacyl-theasaponin E methyl ester in Table 3 represented saponins macromolecules containing five different types saponin aglycone. Therefore, the two wouldn’t be unified, and the latter wouldn’t replace the former.

Description of saponins presence in compost is repeated twice. There is a first sentence “saponin contents (...) primarily consisted of five saponin aglycones of C. oleifera –A, B, C, D, and E- and four of their derivatives.” and below a second sentence “Tea saponins in the compost samples mainly included (...)”, which agrees with the information in tables 3 and 5. I would suggest replacing the first sentence with the second one. In addition, the number of glycosides/aglycones seems to be wrong in one of the sentences, as the first says “5 aglycones/4 derivatives” and the second describes 6 aglycones and 3 glycosides. Please, verify the number of glycosides/aglycones.

Response: Based on your suggestion, the first sentence has been deleted.

Page 10:

(Table 6) instead of (see Table 6).

Response: Similar error had been revised in the whole manuscript.

Is there a direct relationship between ammoniacal nitrogen or nitrate nitrogen and tannin or saponin contents? Or, are the found correlations just due to their increase/decrease with composting time?

Response: NO, there isn’t. Ammoniacal nitrogen or nitrate nitrogen affected the content of tannin or saponin contents by affecting the number of microorganisms. Tannin and saponin contents had direct correlation with composting time. In addition, it had indirect correlations with other parameters of compost through composting time. Because time would affect the changes of other parameters directly.

Page 11

Table 6: Definition of TN, TP and TK should be stated as a footnote.

Response: These had been revised. Definition of TN, TP and TK has been stated as a footnote.

Conclusion

Please check the saponins names (as in abstract and page 6) so they are consistent throughout text and tables. Also, check the number of aglycones and glycosides.

Response: Thanks again for your suggestion. Saponins names and the number of aglycones and glycosides had been checked in whole manuscript. And the wrong part has been corrected. 

Responses to Editor

Thank you for your pointing out problems. All the problems were revised in this revision.

1. According to requisition, minimal data has been uploaded set as a Supporting Information file.

2. Ethics Statement：For experiment and collection locations no specific permits were required for the described field studies because the whole experiment process did not involve endangered or protected plant species or privately-owned locations.

3. Funding-related information has been removed from the manuscript. Please help supplement following funding information. “This work was financially supported by the Provincial Department of Science and Technology of Zhejiang, China, Grant NO.2017C02022.

4. Competing Interests section: The authors have declared that no competing interests exist.

---

## [Editor Report · Decision Letter 1]

24 Feb 2020

PONE-D-19-36037R1

Changes in tannin and saponin components during co-composting of Camellia oleifera Abel shell and seed cake

PLOS ONE

Dear Mrs Zhang,

Thank you for submitting your manuscript to PLOS ONE. After careful consideration, we feel that it has merit but does not fully meet PLOS ONE’s publication criteria as it currently stands. Therefore, we invite you to submit a revised version of the manuscript that addresses the points raised during the review process.

L59: should stand "pH"

29-31: wrong syntax

84: "Composting" instead of "Compost"

Do not capitalize "liquid" in "Liquid chromatography"

Considering the comment by R#2 about the term "tea saponins", I would recommend authors to stick to the reviewer's suggestion and use "*C. oleifera* saponins

I don't see any change was made in the Conclusion section.

We would appreciate receiving your revised manuscript by Apr 09 2020 11:59PM. To enhance the reproducibility of your results, we recommend that if applicable you deposit your laboratory protocols in protocols.io, where a protocol can be assigned its own identifier (DOI) such that it can be cited independently in the future. For instructions see: http://journals.plos.org/plosone/s/submission-guidelines#loc-laboratory-protocols

We look forward to receiving your revised manuscript.

Kind regards,

Branislav T. Šiler, Ph.D.

Academic Editor

PLOS ONE

---

## [Author Response · Author response to Decision Letter 1]

2 Mar 2020

Reviewer #1: The authors gave an intresting work focus on the degradation of tannin and saponin during composting of the shell and seed cake of Camellia oleifera Abel. Which make a lot of sence for making resue of the Camellia oleifera Abel wastes without any harmful effect caused by tannin and saponin. However, the manuscript in current version was not good enough to publish.

Response: Thank you for your working on our manuscript. All the points in this comment were already revised in this revision, and we also made some other changes that were highlighted by red color.

First is introduction. The authors gave enough references to support the idea that the shell and seed cake of Camellia oleifera Abel should be and can be treated by composting, but why the tannin and saponin could be degraded? More details about the research progress about it should be given, and the challenges should be pointed out based these references.

Response: Thank you for your suggestions. These were revised in introduction. Firstly, the reasons for the degradation of tannin and saponin were added in the introduction. Secondly, the biggest challenge was experimental design, because there were no references concerning tannin and saponin of C. oleifera shells and seed cakes. This point has also been added in the introduction. And details about the research progress have been explained in detail in the experimental methods (shown in line56-60). 

Second is the sturcture of the manuscript. For as a composting experiment, its effect should be considered based on a certain condition that the composting was normally proceeded. And then, the changes of nutrients, maturity and degradation of tannin and saponin should be described one by one, and then the ralationship were analyzed.

Response: Your suggestion makes some sense. In this manuscript, the results and discussion of changes of tannin and saponin were analyzed strictly in accordance with the composting normal process. However, the changes of nutrients and maturity were not only the certain condition of changes of tannin and saponin but also the results of changes of tannin and saponin. We were aiming to investigate the changes of tannin and saponin in this work. As shown in results and discussion, combined with the changes of nutrients, maturity and degradation of tannin and saponin were analyzed. Thus, it’s difficult and illogical to change the structure in the order you suggested fully. So, we retained the arrangement. 

The authors gave a same name and contents for the first two parts in results and discussion part, and the discussion on the results was not enough. Both of problem made the results and discussion as a report rather than a research paper.

Response: First two parts in results and discussion part analyzed and discussed the changes of tannin and saponin from content and composition, respectively. The titles of the two parts were too similar that it was hard to distinguish, so they were merged. And results and discussion part supplemented the discussion.

The last is the strange figures and tables. Both of the figures and tables were not shown as standard ones in any publication.

1) the figure numbers were not consistent with those in the manuscript.

2) the contents of tannin and saponin were shown in nearly all of the figures, which should be discussed in the most related one of them or shown separately.

3) the words were shown in different format in various figures.

Response: Thank you for your pointing out the problem. All the formatted problems about figures and tables were revised in this revision. The contents of tannin and saponin were shown in all the figures to compare the impact of the same compost parameters on them. In addition, it also enriched the information of the graph. In results and discussion part, the effects of same compost parameters on tannin and saponin were discussed separately.

Responses to Reviewer 2:

Reviewer #2: The article deals with co-composting of Camellia oleifera shells and seed cake, which are by-products obtained during the oil extraction from Camellia oleifera. The work provides relevant information on plant nutrition and safety parameters using Camellia oleifera compost, which is interesting from the agronomic and environmental point of view. The valorization of oil industry by-products is a hot topic, as they are an environmental concern, and their reuse would improve the sustainability of the oil production system. Therefore, I suggest to accept the article for its publication after some amendments of the text.

Response: Thank you for your working on our manuscript. All the points in this comment were already revised in this revision, and we also made some other changes that were highlighted by red color.

Abstract

Replace the text “(...) Camellia oleifera Abel. It was a family of theaceae, small evergreen trees” with “(...) Camellia oleifera Abel, which is a small evergreen tree in the family Theaceae.”

Correct the sentence “were found low in content”: four small-molecule phenolic acids were found in low contents.

Correct the verb tense “it takes 60 days”: it took 60 days.

The sentence “tannin and saponin contents were examined for their correlations (...) repeats a previous sentence in the abstract. It would be simpler and clearer as follows: “Tannin and saponin correlations with the C/N ratio, germination index (GI), and the Solvita maturity

Response: Thank you for your suggestions. These sentences were revised according to your suggestions, and the tense in the whole manuscript was checked (shown in 22-23, 26, 29-30).

Page 2 and beyond

Check the bibliography format, most of the citations are undistinguished numbers within the text.

Response: This format was according to the requirement of PLOS ONE’s bibliography format (for example, “[1]” or “[2-5]”or “[3,7,9]”).

Page 4

There is a reference error message (analytical methods).

The year of the reference 8 is wrong: Savolainen, H. (1992). Tannin content of tea and coffee. Journal of Applied Toxicology, 12(3), 191-192.

Typographical error: pro-gram.

Response: This have been revised, and checked for other bibliographic information. 

Page 5

Table 6 is referred after table 3, therefore, it should be table 4. The rest of table numbers should be corrected accordingly.

Response: This was because Table 6 summarized the correlation analysis results of all indicators of compost and tannin and saponin contents. Hence, Table 6 was referred in here. However, the order of Tables was correct.

Please add a coma and reorganize the sentence for better understanding: “As composting proceeded, the content of every phenolic acid compound declined, except for ellagic acid”.

Response: This sentence had been revised in the text in easy understanding way (shown in 142-146).

Page 6

Tables 2 and 3. Is response intensity the relative response in the MS detector? It should be stated as a table footnote.

Response: No, response intensity is the relative response in the Liquid chromatography (LC), it has been stated as a table footnote.

The term “tea saponins” is confusing, as the article does not deal with tea (Camellia sinensis). I would refer it as C. oleifera saponins or its common name (tea oil camellia, provided that it has been defined previously in the introduction).

Response: “C. oleifera saponins” replaced term “tea saponins” and has been explained in detail in 166-167 lines on page 7 in revision. 

There is a confusion with saponin aglycones names. In some parts of the text (e.g. abstract) they are defined as “A, B, C, D and E, while in others (and table 3) as theasapogenol A, camelliagenin A, camelliagenin B, camelliagenin C, camelliagenin D and camelliagenin D derivative. Only one name should be used, and I would suggest using the last ones, as they are more descriptive of the compounds.

Response: In the introduction, A, B, C, D, and E all represented saponin aglycone, but theasapogenol A, camelliagenin A, camelliagenin B, camelliagenin C, camelliagenin D and camelliagenin D derivative, and desacyl-theasaponin E methyl ester in Table 3 represented saponins macromolecules containing five different types saponin aglycone. Therefore, the two wouldn’t be unified, and the latter wouldn’t replace the former.

Description of saponins presence in compost is repeated twice. There is a first sentence “saponin contents (...) primarily consisted of five saponin aglycones of C. oleifera –A, B, C, D, and E- and four of their derivatives.” and below a second sentence “Tea saponins in the compost samples mainly included (...)”, which agrees with the information in tables 3 and 5. I would suggest replacing the first sentence with the second one. In addition, the number of glycosides/aglycones seems to be wrong in one of the sentences, as the first says “5 aglycones/4 derivatives” and the second describes 6 aglycones and 3 glycosides. Please, verify the number of glycosides/aglycones.

Response: Based on your suggestion, the first sentence has been deleted.

Page 10:

(Table 6) instead of (see Table 6).

Response: Similar error had been revised in the whole manuscript.

Is there a direct relationship between ammoniacal nitrogen or nitrate nitrogen and tannin or saponin contents? Or, are the found correlations just due to their increase/decrease with composting time?

Response: NO, there isn’t. Ammoniacal nitrogen or nitrate nitrogen affected the content of tannin or saponin contents by affecting the number of microorganisms. Tannin and saponin contents had direct correlation with composting time. In addition, it had indirect correlations with other parameters of compost through composting time. Because time would affect the changes of other parameters directly.

Page 11

Table 6: Definition of TN, TP and TK should be stated as a footnote.

Response: These had been revised. Definition of TN, TP and TK has been stated as a footnote.

Conclusion

Please check the saponins names (as in abstract and page 6) so they are consistent throughout text and tables. Also, check the number of aglycones and glycosides.

Response: Thanks again for your suggestion. Saponins names and the number of aglycones and glycosides had been checked in whole manuscript. And the wrong part has been corrected. 

Responses to Editor

Thank you for your working on our manuscript. All the points in this comment were already revised in this revision, and we also made some other changes that were highlighted by red color.

L59: should stand "pH"

Response: This have been revised, and checked for the whole manuscript.

29-31: wrong syntax

Response: This have been revised.

84: "Composting" instead of "Compost"

Response: This have been revised.

Do not capitalize "liquid" in "Liquid chromatography"

Response: This have been revised.

Considering the comment by R#2 about the term "tea saponins", I would recommend authors to stick to the reviewer's suggestion and use "C. oleifera saponins

Response: "C. oleifera saponins" replaced "tea saponins" in the whole manuscript according to reviewer's suggestion.

I don't see any change was made in the Conclusion section.

Response: Conclusion section has been simplified.

Conclusion as below:

 Tannins and saponins of C. oleifera shell and seed cake can be degraded, and the content of NO3-N can be increased by composting technology with 60 days. Thus, qualified compost products that are beneficial to plant growth can be obtained. In the experimental compost samples, the identified tannins mainly consisted of 11 phenolic acid compounds, but four of these were small-molecule phenolic acids relatively low in content. The saponins identified mainly consisted of five saponin aglycones of C. oleifera (named A, B, C, D, and E) and four of their derivatives. With the progress of composting, the content of tannins and saponins gradually decreased to a safe range. Because microbially secreted enzymes converted most large-molecule phenolic acid compounds to small-molecule compounds and their derivatives, and saponins decomposed to saponin aglycones. In addition, the tannin and saponin contents were inversely related to the TN, TP, TK, GI, solvita maturity index and total nutrient contents, and the negative correlation with solvita maturity index was significant. Yet the tannin and saponin contents were significantly positively correlated with C/N and content of NO3-N.

---

## [Editor Report · Decision Letter 2]

4 Mar 2020

Changes in tannin and saponin components during co-composting of Camellia oleifera Abel shell and seed cake

PONE-D-19-36037R2

Dear Dr. Zhang,

We are pleased to inform you that your manuscript has been judged scientifically suitable for publication and will be formally accepted for publication once it complies with all outstanding technical requirements.

With kind regards,

Branislav T. Šiler, Ph.D.

Academic Editor

PLOS ONE
---

## [Editor Report · Acceptance letter]

9 Mar 2020

PONE-D-19-36037R2 

Changes in tannin and saponin components during co-composting of *Camellia oleifera* Abel shell and seed cake 

Dear Dr. Zhang:

I am pleased to inform you that your manuscript has been deemed suitable for publication in PLOS ONE. Congratulations! Your manuscript is now with our production department. 

With kind regards,

on behalf of

Dr. Branislav T. Šiler 

Academic Editor

PLOS ONE